# Validation of a Food Frequency Questionnaire: VioScreen-Allergy

**DOI:** 10.3390/nu16213772

**Published:** 2024-11-02

**Authors:** Kaci Pickett-Nairne, Deborah Glueck, Jessica Thomson, Rick Weiss, Kelly N. Z. Fuller, Stefka Fabbri, Claudia Schaefer, Courtney Evans, Emily Bowhay, Monica Martinez, Wendy Moore, David Fleischer, Carina Venter

**Affiliations:** 1Department of Pediatrics, Children’s Hospital Colorado, University of Colorado School of Medicine, Aurora, CO 80045, USA; kaci.pickett-nairne@cuanschutz.edu (K.P.-N.); deborah.glueck@cuanschutz.edu (D.G.); claudia.schaefer@cuanschutz.edu (C.S.); 2Delta Human Nutrition Research Program, USDA Agricultural Research Service, Stoneville, MS 38776, USA; jessica.thomson@usda.gov; 3Viocare, Inc., Princeton, NJ 08542, USA; weiss@viocare.com (R.W.); fullmeyer@viocare.com (K.N.Z.F.); 4Obstetrics & Gynecology, Denver Health Hospital, Denver, CO 80204, USA; stefka.fabbri@dhha.org (S.F.); monica.martinez@dhha.org (M.M.); 5Colorado Child Health Research Institute, Children’s Hospital Colorado, Aurora, CO 80045, USA; courtney.evans@childrenscolorado.org (C.E.); emily.bowhay@childrenscolorado.org (E.B.); wendy.moore2@childrenscolorado.org (W.M.); 6Section of Pediatric Allergy and Immunology, Children’s Hospital Colorado, University of Colorado, Boulder, CO 80309, USA

**Keywords:** food frequency questionnaire, intermethod reliability, test–retest reliability, external validity, maternal diet index

## Abstract

Background/Objectives: An adapted version of an online pictorial food frequency questionnaire (FFQ), VioScreen-Allergy, assesses total dietary intake and intake of allergens and foods in the maternal diet index (MDI), linked to offspring allergy. This study assessed intermethod reliability, test–retest reliability, and external validity of the VioScreen-Allergy. Methods: Females of childbearing age were recruited at Denver Health and Children’s Hospital, Colorado, USA, and were asked to complete four 24 h recalls and two VioScreen-Allergy FFQs over the course of a month. All those with at least two 24 h dietary recalls and both VioScreen-Allergy assessments were analyzed. Energy-adjusted and non-adjusted linear mixed models (1) compared MDI scores and intake of nutrients and allergens as measures of intermethod reliability; (2) evaluated VioScreen-Allergy test–retest reliability as differences between repeated measurements; and (3) assessed external validity by modeling associations between VioScreen-Allergy-derived intake of beta-carotene and orange vegetables and Veggie Meter^®^-assessed skin carotenoids. Bonferroni corrections controlled multiple comparisons within the assessment. Results: Of 53 participants enrolled, 25 demographically dissimilar participants were included in the analysis. There were no significant differences between 24 h recall and VioScreen-Allergy mean intakes of macronutrients, micronutrients, allergens, or MDI, except for Vitamin C, niacin, and cashew allergen protein. There were no significant differences between repeated measurements of VioScreen-Allergy, either energy-adjusted or unadjusted. Both beta-carotene and orange vegetable servings were significantly associated with Veggie Meter^®^. Conclusions: Although non-significance could have been due to low power, clinical as well as statistical assessments of intermethod reliability, test–retest reliability, and external validity suggest that VioScreen-Allergy has reasonable utility for trials assessing food allergens and MDI in the context of overall intake. The VioScreen questionnaire can also be used in future studies to assess macro- and micronutrient intake. Additional validation studies assessing different portion sizes and foods eaten by infants and young children are currently undergoing.

## 1. Introduction

The World Allergy Organization [1] and the Institute of Medicine [2] state that the prevalence of atopic dermatitis and food allergies is rising dramatically. Our team has demonstrated that a maternal diet during pregnancy rich in vegetables and yogurt and with reduced intakes of red meat, cold cereal, fried potatoes, rice and grains, and 100% fruit juice was associated with reduced odds of atopic dermatitis, asthma, allergic rhinitis, and wheeze in offspring by four years of age [3]. The next step is to test whether a diet of this sort reduces offspring allergy risk in a randomized controlled trial (RCT).

One limitation of current RCTs seeking to change diet during pregnancy to reduce risk of offspring allergy is that no study has used validated measures to collect data on dietary intake [4]. However, both validated and unvalidated food frequency questionnaires (FFQs) have been used in many observational studies focusing on maternal diet intake during pregnancy and offspring allergy outcomes [4]. The limitations of these FFQs are that they are not validated to measure intake of food allergens other than by assessing frequency of intake [5]. No questionnaire has been validated to measure the maternal diet index (MDI). Validated food recall measures, such as the ASA-24 [6] and the Diet History Questionnaire (DHQ) [7], do not clearly distinguish between different forms of nuts and seeds eaten, a critical need for allergen assessment. This poses a practical dilemma for studies where measuring consumption of specific food allergens is crucial.

Large scale population-based studies often include an extensive number of data collection tools. To reduce participant burden, the time used to complete dietary questionnaires should be kept to a minimum [8,9]. One rationale for using FFQs is that the respondent burden is lighter than when repeated food diaries or 24 h recalls need to be completed, leading to higher response rates [10]. In addition, FFQs do not require interviewers. Because FFQ looks backwards in time, customary eating habits are not influenced and potentially changed [10].

However, FFQs can have several weaknesses, including incorrect recall and errors in the approximate quantification of food intake. Thus, validation of FFQs is crucial [10]. As an improvement to paper-based FFQs, more sophisticated online forms of FFQs, such as the VioScreen FFQ (Viocare), use complex skip algorithms to reduce participant burden and errors. VioScreen is a unique self-administered web and mobile questionnaire that incorporates over 1200 food images used by patients to select foods they consume with details on frequency, portion size, and preparation. The VioScreen is currently being used by the American Gut Study [11].

Because the originally validated VioScreen FFQ [12] did not include assessment of allergen intake, one of the authors, Dr. Venter, has worked closely with the VioScreen team to add questions that enable assessment of allergens. This newly adapted version of VioScreen, VioScreen-Allergy, has not been validated for overall intake, for intake of food allergens, or for scoring of the MDI [13].

Ideally, FFQs such as the VioScreen would be compared with gold-standard forms of food recalls and a biomarker [14]. Weighted-food records or written diet records could be considered the best methods for validating food frequency questionnaires. However, 24 h recalls are less demanding, depend less on the literacy of study participants, and are more commonly used in both clinical and research settings [10].

The goals of this present work are to (1) compare MDI scores and intake of nutrients and allergens as measures of intermethod reliability between 24 h recall and VioScreen-Allergy; (2) evaluate VioScreen-Allergy test–retest reliability as differences between repeated measurements; and (3) assess external validity by modeling associations between VioScreen-Allergy-derived intakes of beta-carotene and orange vegetables and Veggie Meter^®^-assessed skin carotenoids.

## 2. Materials and Methods

### 2.1. Study Design

We performed a longitudinal, observational cohort study (with recruitment between November 2023 and April 2024) at the prenatal clinic of Denver Health and from Children’s Hospital Colorado, both in Colorado, USA. The study was approved by the Colorado Multiple Institutional Review Board (COMIRB) as study number 22-0904.

### 2.2. Inclusion and Exclusion Criteria

Participants were included if they were (1) between 18 and 40 years of age and (2) reported being assigned female at birth. Participants were excluded if they were pregnant more than 26 weeks gestation, were experiencing limited or reduced food intake for any reason, were unwilling to participate in the study due to time commitment or other reasons, or had diminished capacity for study compliance due to conditions such as active untreated substance use disorder, untreated/uncontrolled schizophrenia, bipolar disorder, or active psychosis. All participants gave written informed consent prior to enrollment and study procedures.

### 2.3. Study Questionnaires

Study procedures are indicated in Table 1. Links to the baseline demographic questionnaire, a 24 h dietary recall that is directly linked to the Nutrition Data system for Research (NDSR) [15], and the VioScreen-Allergy online self-administered questionnaire [11] were emailed to the study participants at baseline and a month after baseline. The baseline questionnaire collected demographic data, including age, relationship status, pregnancy status, highest level of education, and total household income in the last year. Study participants were called by the study dietitian to complete up to four x 24 h dietary recalls, one week apart. Participants were compensated for their participation.

### 2.4. Veggie Meter^®^ Biomarker

The Veggie Meter^®^ is a non-invasive portable device that uses spectroscopy to measure skin carotenoid levels, a concentration biomarker of usual fruit and vegetable intake [16].

The Veggie Meter^®^ test was carried out during two visits: one at baseline and one month later. To perform the Veggie Meter^®^ test, the index finger is placed in the Veggie Meter^®^, which is similar to a pulse oximeter. The readings are available in seconds and taken in duplicate.

### 2.5. Maternal Diet Index

The MDI [17] was computed using data regarding maternal diet during pregnancy. The index includes dietary components associated with allergy prevention (yogurt, vegetables) and dietary components associated with an increased risk of allergy (red meat, rice/grains, fried potatoes, cold cereals, 100% fruit juice). Higher MDI scores indicate a more allergy preventive diet (increased intake of yogurt and vegetables; reduced intake of red meat, rice/grains, fried potatoes, cold cereals, 100% fruit juice), while lower MDI scores represent a less allergy-preventive diet (reduced intake of yogurt and vegetables; increased intake of red meat, rice/grains, fried potatoes, cold cereals, 100% fruit juice). The theoretical range of MDI scores is 0 to 100 [17]. Coding for the MDI is freely available for download from https://github.com/CarinaVenter/MaternalDietIndex (accessed on 28 October 2024). The code allows a researcher to compute the maternal diet index using either SAS Version 9.4 (SAS Institute, Cary, NC, USA) or R statistical software v 4.4.1.

The MDI was computed as described in Venter et al. [17]. The MDI is a weighted sum of foods associated with a reduced risk of offspring allergy and foods associated with an increased risk of offspring allergy. Foods associated with a decreased risk of offspring allergy included yogurt and vegetables. Foods associated with an increased risk of offspring allergy included red meat, cold cereal, fried potatoes, rice and grains, and 100% fruit juice.

### 2.6. Statistical Analysis

Statistical analyses were performed using R software v. 4.4.1, with linear mixed models implemented in SAS Version 9.4. Demographic baseline characteristics of the cohort were reported as N (%). Unadjusted and energy-adjusted general linear mixed models compared (1) intake derived from 24 h dietary recalls and the VioScreen-Allergy and (2) repeated measures of VioScreen-Allergy as a measure of test–retest reliability. The approach accounted for potentially different numbers of measures of 24 h recalls (number of 24 h recalls conducted= 2–4) completed by each study participant. An unstructured covariance was used to account for the in-person correlation of the diet measures. Wald tests with Kenward-Roger [18] degrees of freedom assessed significance. Macronutrients were reported in grams and as percent of energy intake. To control multiple comparisons within each category (i.e., macronutrients, micronutrients, and allergens), we used a Bonferroni approach. Macronutrients were compared with an alpha level of 0.05/8 = 0.0063, micronutrients to an alpha level of 0.05/14 = 0.0036, and allergens to an alpha level of 0.05/7 = 0.007. All other comparisons used a Type I error rate of 0.05. A general linear univariate model assessed associations between (1) intake of orange vegetables and (2) beta-carotene intake with skin carotenoid measures from the Veggie Meter^®^. Regression (beta) coefficients, 95% confidence intervals, and *p*-values were reported for each comparison.

## 3. Results

### 3.1. Demographic Information

Of the 53 people who began the study, 25 study participants completed at least two 24 h dietary recalls and both VioScreen-Allergy assessments. Those included in the analysis were significantly older, more likely to be married, and less likely to be pregnant than those not included in the analysis (Appendix A; Table 2).

### 3.2. Intermethod Reliability

There were no significant differences [mean difference, (95% confidence interval)] between averages of 24 h dietary recalls and VioScreen-Allergy energy [96.7 kCal, (−103.9, 297.4)], energy adjusted macronutrients: fat [−0.9 g, (−5.9, 4.0)], protein [3.5 g, (−2.1, 9.2)], carbohydrates [−1.9 g, (−15.6, 11.9)], or in any adjusted micronutrient, except Vitamin C [−49.3 mg, (−68.7, −29.9)] and niacin [−15.1 mg niacin equivalent, (−17.7, −12.5)] (Table 3). Intakes of allergens were similar, except for cashew protein [−0.4 g (−0.7, −0.2)] (Table 3). The MDI did not differ significantly between modalities [0.5 units (−0.2, 1.2)] (Table 3).

### 3.3. External Validity

Skin carotenoid measures from the Veggie Meter^®^ had a mean value of 339 (standard deviation (sd) = 121). Beta-carotene intake (Table 4) derived from VioScreen-Allergy was significantly positively associated with skin carotenoid measures from the Veggie Meter^®^ (beta = 0.017, 95% CI = (0.005, 0.029, *p* = 0.008), as was orange vegetable servings (beta = 255, 95% CI = (68, 442), *p* = 0.01).

### 3.4. Test–Retest Reliability

Table 5 shows tests of differences between the first and second VioScreen-Allergy measures for macronutrients, micronutrients, allergen intake, and the MDI. There were no significant differences between repeated measurements of VioScreen-Allergy, either adjusted or unadjusted.

## 4. Discussion

This study set out to test intermethod reliability between repeated 24 h dietary recalls and a novel online, pictorial questionnaire, VioScreen-Allergy, measure test–retest reliability of VioScreen-Allergy, and assess external validity by comparing associations between measures of orange vegetable and beta-carotene intake obtained from the VioScreen-Allergy against the Veggie Meter^®^. Our results indicated no significant differences between macronutrients, micronutrients, allergen intake, and the MDI assessed via 24 h recalls and VioScreen-Allergy, except for Vitamin C, niacin, and cashew allergen protein. There were no significant differences between repeated measurements of VioScreen-Allergy, either energy-adjusted or unadjusted. Both beta-carotene and orange vegetable servings were significantly associated with Veggie Meter^®^ readings.

The approach used in the study to assess intermethod reliability, test–retest reliability, and external validity was to fit unadjusted and energy-adjusted general linear mixed models. The approach had several advantages and several weaknesses. An advantage of the approach was the ability to estimate mean differences between methods and repeated measures and thus assess both clinical and statistical significance. The approach also accounted for correlation between repeated measures and for differences in variance incurred by having potentially different numbers of 24 h recalls between study participants. As we reported mean differences rather than Pearson and Spearman correlations, we were unable to directly contextualize our approach compared with the most relevant previously published study. However, although we could not compare the statistics directly, the direction of the results was similar. The previous study by Kristal et al. [12] compared the VioScreen FFQ to repeated 24 h recalls and found similarly intermethod reliability between the 24 h recalls and the VioScreen for fat, carbohydrate, and protein.

The validated VioScreen questionnaire has been used in a wide range of studies focusing on dietary intake in individuals with diseases such as cancer [19,20], pancreatitis [21,22], and Parkinson’s disease [13], as well as studies assessing dietary intake during pregnancy [23] and healthy eating patterns in individuals without disease [24,25]. While the VioScreen questionnaire has not previously been used in studies focusing on allergy outcomes, we are planning future studies to do just that.

Results from a systematic review indicated that skin carotenoid measures are valid biomarkers of fruit and vegetable intake [26]. The Veggie Meter^®^ and assessment of skin carotenoid levels have been used in dietary intake studies [27,28,29] and for studying associations between dietary intake and health outcomes [30,31,32]. More specifically, a study found significantly higher levels of skin carotenoids measured using the Veggie Meter^®^ in children with cow’s milk allergy compared with children without cow’s milk allergy [33].

Other studies assessing the intermethod reliability of FFQs against other standards showed energy-adjusted, deattenuated intermethod reliabilities for macronutrients of 0.41, 0.51, and 0.41 for protein, carbohydrate, and total fat, respectively, using the Block questionnaire [34]. Again, these results are similar to the intermethod agreement reported by us. Beasley et al. [35] reported that the intermethod reliability of energy-adjusted but not deattenuated macronutrients from the computer-administered Diet History Questionnaire (DHQ) was again and positive, similar to the results reported by us. Correlations of the DHQ with 24 h recall were 0.45, 0.38, and 0.30 for protein, carbohydrate, and total fat, respectively. For micronutrient intake, again, our results were similar in direction to previously published work.

Our approach for assessing test–retest reliability was to compare repeated measures of the Vioscreen-Allergy with each other. Such an approach reflects an assumption that dietary intake overall does not change much over one month of time when no intervention is delivered. The veracity of this assumption is demonstrated by Kristal et al. [12], who showed that the correlations between the two administrations of the VioScreen (test–retest reliability) for fat, carbohydrate, and protein were 0.60, 0.63, and 0.73. Our data also showed remarkable similarity between repeated measures of the VioScreen-Allergy. There is currently only one validated FFQ that reported on food allergen intake by Venter et al. [5]. In this study, the authors reported that for intermethod reliability, agreement was above 90% for intake of milk, wheat, and white fish.

Our data showed sufficient test–retest reliability, except for cashew allergen protein intake, a rare exception that may reflect differences in episodic intake of nuts from month to month. This is one of the main reasons why 24 h recalls are not considered an adequate dietary intake tool for foods consumed infrequently [36]. For test–retest reliability, agreement between responses to general questions on the two reliability questionnaires was higher than 75% for oily fish, peanut, wheat, shell fish, and egg in the study by Venter et al. [5]. Our data also showed sufficient test–retest reliability, both unadjusted and adjusted for energy intake.

Our validation against a biomarker showed significant positive linear association between orange vegetable intake and Veggie Meter^®^ skin carotenoids and significant positive linear association between beta-carotene intake and Veggie Meter^®^ skin carotenoids. These findings again mirror the direction of other studies. A previous study comparing FFQ data against the Veggie Meter^®^ showed children’s Veggie Meter readings were significantly correlated with parent-reported child total fruit and vegetable frequency score, τ = 0.16 (*p* = 0.04) [26]. Another study by Amoah et al. [37] showed positive correlations of around 0.69 between servings of carotenoid-rich foods and the carotenoid reflection scores. Finally, we were able to show sufficient test–retest reliability of the MDI, a finding that no other paper has reported.

Limitations of our study include a small sample size, which may have attenuated power. However, careful examination of the mean differences in intake observed between and within the methods suggests that the differences are both clinically and statistically insignificant. Generalizability may be limited as the majority of participants were highly educated and had a high household income.

## 5. Conclusions

In conclusion, we have provided evidence supporting the use of VioScreen-Allergy as a dietary data collection tool. VioScreen-Allergy can fill the need for validated dietary questionnaires that can measure overall dietary intake while collecting detailed information on food allergen intake. The use of VioScreen-Allergy will enable us to measure the MDI while conducting randomized controlled trials. It will also enable careful dietary intake measurements during future observational studies. The VioScreen questionnaire can also be used in future studies to assess macro- and micronutrient intake. Additional validation studies assessing different portion sizes and foods eaten by infants and young children are currently undergoing.

## Figures and Tables

**Table 1 nutrients-16-03772-t001:** Overview of study procedures.

	Baseline Visit (Week 1)	Week 2	Week 3	Week 4	Week 5	Week 6 (End of Study)
**Visit Location**	**Clinic**	**Phone**	**Phone**	**Phone**	**Phone**	**Clinic**
Informed Consent	**X**					
Baseline questionnaire	**X**					
VioScreen questionnaire	**X**					**X**
Veggie Meter^®^	**X**					**X**
24 h dietary recall		**X ***	**X ^**	**X ***	**X ^**	

X conducted at this time point * Weekend dietary recall. ^ Weekday dietary recall.

**Table 2 nutrients-16-03772-t002:** Demographic and baseline characteristics of cohort.

	Overall (N = 25)
**Age (yrs)** Median (Q1, Q3)	32.3 (28.2, 37.3)
**What is your current relationship status?**	
Married	15 (60%)
Unmarried	10 (40%)
**Pregnant?**	1 (4%)
**What is the highest level of school you have completed?**	
Associate degree	1 (4%)
Bachelors degree	17 (68%)
Masters degree	6 (24%)
Professional or doctorate degree	1 (4%)
**What was your total household income in the last year?**	
Less than $40,000	1 (4%)
$40,000 to $74,999	6 (24%)
$75,000 or more	14 (56%)
Prefer not to answer	4 (16%)

**Table 3 nutrients-16-03772-t003:** Mixed model estimates for the comparison of all available FFQ and VioScreen-Allergy with 95% confidence intervals.

Nutrient	24 h Dietary Recall	VioScreen-Allergy	Difference (Recall vs. VioScreen-Allergy)	*p* Value ^a^	Energy Adjusted Difference (Recall vs. VioScreen-Allergy)	*p* Value ^a^
Energy (kcal)	1626.0 (1503.0, 1749.1)	1529.3 (1370.8, 1687.8)	96.7 (−103.9, 297.4)	0.34		
Total fat (g)	69.7 (63.1, 76.3)	66 (57.5, 74.5)	3.7 (−7.1, 14.4)	0.50	−0.9 (−5.9, 4.0)	0.71
% Energy	37.9 (36.3, 39.5)	38.2 (36.1, 40.2)	−0.3 (−2.9, 2.4)	0.84		
Saturated fat (g)	23.4 (20.7, 26.0)	21.3 (17.9, 24.7)	2.1 (−2.2, 6.4)	0.34	0.4 (−2.2, 3.0)	0.75
% Energy	12.5 (11.7, 13.3)	12.3 (11.2, 13.3)	0.2 (−1.1, 1.6)	0.75		
Polyunsaturated fat (g)	15.2 (13.6, 16.9)	14.3 (12.2, 16.4)	0.9 (−1.8, 3.6)	0.50	0.03 (−1.9, 2.0)	0.97
% Energy	8.5 (7.9, 9.2)	8.2 (7.4, 9)	0.3 (−0.7, 1.4)	0.54		
Monounsaturated fat (g)	24.6 (22.0, 27.3)	24.3 (20.9, 27.7)	0.3 (−3.9, 4.6)	0.87	−1.3 (−3.9, 1.2)	0.30
% Energy	13.3 (12.5, 14.1)	14.1 (13.1, 15.2)	−0.8 (−2.1, 0.5)	0.21		
Carbohydrate (g)	173.8 (158.3, 189.3)	165.4 (145.5, 185.4)	8.4 (−16.9, 33.7)	0.51	−1.9 (−15.6, 11.9)	0.79
% Energy	43.4 (41.4, 45.3)	43 (40.5, 45.5)	0.4 (−2.8, 3.6)	0.82		
Protein (g)	70.8 (65.1, 76.5)	63.7 (56.3, 71)	7.1 (−2.2, 16.4)	0.13	3.5 (−2.1, 9.2)	0.22
% Energy	17.7 (16.7, 18.6)	17 (15.8, 18.2)	0.7 (−0.9, 2.3)	0.38		
Dietary fiber (g)	17.9 (16.0, 19.8)	18.9 (16.4, 21.4)	−1 (−4.1, 2.1)	0.54	−1.8 (−4.4, 0.8)	0.17
EPA ^b^ and DHA ^c^ (g)	1.6 (1.4, 1.8)	1.4 (1.2, 1.7)	0.2 (−0.2, 0.5)	0.39	0.07 (−0.2, 0.4)	0.64
Retinol (µg RE ^d^)	302.9 (259.1, 346.7)	348.2 (291.8, 404.6)	−45.3 (−116.7, 26.1)	0.21	−61.8 (−124.9 1.4)	0.06
Beta carotene (µg RE)	3299.7 (2399.4, 4200.1)	3968.7 (2808.7, 5128.6)	−668.9 (−2137.3, 799.4)	0.37	−778.7 (−2240.0, 682.6)	0.29
Vitamin C (mg)	61.0 (48.2, 73.7)	106.4 (89.9, 122.9)	−45.5 (−66.3, −24.6)	<0.0001	−49.3 (−68.7, −29.9)	<0.0001
Vitamin D (µg)	3.6 (2.7, 4.5)	4.8 (3.7, 6.0)	−1.2 (−2.7, 0.3)	0.11	−1.46 (−2.9, −0.1)	0.04
Niacin (mg NE ^e^)	19 (16.7, 21.4)	32.8 (29.7, 35.8)	−13.7 (−17.5, −9.9)	<0.0001	−15.1 (−17.7, −12.5)	<0.0001
Thiamin (mg)	1.4 (1.3, 1.6)	1.1 (0.9, 1.3)	0.3 (0.0, 0.6)	0.03	0.2 (0.0, 0.5)	0.05
Riboflavin (mg)	1.6 (1.4, 1.7)	1.7 (1.5, 1.9)	−0.1 (−0.4, 0.1)	0.24	−0.2 (−0.4, −0.1)	0.01
Vitamin B-6 (mg)	1.5 (1.3, 1.7)	1.7 (1.5, 2.0)	−0.2 (−0.5, 0.1)	0.14	−0.3 (−0.5, −0.1)	0.02
Vitamin B-12 (µg)	3.1 (2.7, 3.6)	3.9 (3.2, 4.5)	−0.7 (−1.5, 0.1)	0.07	−0.9 (−1.6, −0.2)	0.01
Folate (µg FE ^f^)	321.7 (291.4, 352.1)	295.8 (256.7, 334.9)	25.9 (−23.5, 75.4)	0.30	10.6 (−27.6, 48.8)	0.58
Calcium (mg)	764.4 (680.8, 847.9)	858.8 (751.2, 966.4)	−94.4 (−230.7, 41.8)	0.17	−138.8 (−240.0, −37.6)	0.01
Iron (mg)	11.8 (10.6, 12.9)	9.8 (8.4, 11.3)	1.9 (0.1, 3.8)	0.04	1.3 (−0.1, 2.7)	0.07
Zinc (mg)	9 (8.1, 9.9)	8.4 (7.2, 9.6)	0.6 (−0.9, 2.1)	0.44	0.1 (−1.0, 1.2)	0.85
Almond protein (g)	0.7 (0.3, 1.0)	1.1 (0.7, 1.5)	−0.4 (−1.0, 0.1)	0.09	−0.5 (−0.98, 0.04)	0.07
Cashew protein (g)	0.1 (0.0, 0.2)	0.5 (0.3, 0.7)	−0.4 (−0.7, −0.2)	0.0004	−0.4 (−0.7, −0.2)	0.0003
Egg protein (g)	3.1 (2.2, 4.1)	3.2 (2.0, 4.4)	0 (−1.6, 1.5)	0.95	−0.2 (−1.7, 1.3)	0.76
Milk protein (g)	13.3 (11.0, 15.6)	16.2 (13.2, 19.2)	−2.9 (−6.6, 0.9)	0.14	−3.7 (−7.1, −0.4)	0.03
Peanut protein (g)	2.5 (1.3, 3.8)	3.3 (1.7, 4.9)	−0.8 (−2.8, 1.2)	0.44	−1.0 (−3.0, 0.9)	0.30
Sesame protein (g)	0.1 (0.0, 0.1)	0.1 (0.1, 0.2)	−0.1 (−0.1, 0.0)	0.24	−0.1 (−0.2, 0.04)	0.23
Walnut protein (g)	0 (0.0, 0.1)	0.1 (0.0, 0.1)	0 (−0.1, 0.0)	0.43	0 (−0.1, 0.0)	0.41
Overall MDI ^g^	73.8 (73.4, 74.2)	73.3 (72.8, 73.8)	0.5 (−0.1, 1.2)	0.13	0.5 (−0.2, 1.2)	0.13

^a^ Bonferroni adjusted significance levels = 0.0063 for macronutrients, 0.0036 for micronutrients, and 0.007 for allergens. ^b^ EPA = eicosapentaenoic acid. ^c^ DHA = docosahexaenoic acid. ^d^ RE = retinol equivalents. ^e^ NE = niacin equivalents. ^f^ FE = folate equivalents. ^g^ MDI = Maternal Diet Index.

**Table 4 nutrients-16-03772-t004:** Linear regression between vegetable and nutrient intake from baseline VioScreen-Allergy (predictor) and veggie meter readings (outcome).

Component	Mean (sd)	Beta Estimate (95% CI)	*p*-Value
Beta carotene (µg RE)	4484 (3718)	0.017 (0.005, 0.029)	0.008
Orange Vegetable Servings	0.18 (0.24)	255 (68, 442)	0.01

**Table 5 nutrients-16-03772-t005:** Mixed model estimates and 95% confidence intervals for the test–retest reliability comparison of selected nutrients from the VioScreen-Allergy.

Nutrient	VioScreen-Allergy 1	VioScreen-Allergy 2	Difference (VioScreen-Allergy 1 vs. 2)	*p* Value ^a^	Energy Adjusted Difference (VioScreen-Allergy 1 vs. 2)	*p* Value ^a^
Energy (kcal)	1701.3 (1143.9, 2258.7)	1357.3 (799.9, 1914.7)	344.0 (−444.3, 1132.3)	0.39		
Total fat (g)	72.8 (47.8, 97.7)	59.3 (34.3, 84.2)	13.5 (−21.8, 48.8)	0.45	−1.5 (−9.5, 6.4)	0.71
% Energy	37.7 (25.4, 50.0)	38.6 (26.3, 50.9)	−0.9 (−18.3, 16.5)	0.92		
Saturated fat (g)	23.4 (14.6, 32.2)	19.2 (10.4, 28.0)	4.2 (−8.3, 16.6)	0.51	−0.9 (−5.2, 3.3)	0.66
% Energy	12.1 (7.9, 16.3)	12.4 (8.2, 16.7)	−0.3 (−6.3, 5.6)	0.92		
Polyunsaturated fat (g)	16.1 (10.5, 21.8)	12.5 (6.8, 18.1)	3.7 (−4.3, 11.7)	0.36	0.5 (−2.6, 3.6)	0.76
% Energy	8.3 (5.4, 11.3)	8.1 (5.2, 11.0)	0.2 (−3.9, 4.4)	0.91		
Monounsaturated fat (g)	26.5 (17.4, 35.6)	22.1 (13.0, 31.2)	4.4 (−8.5, 17.3)	0.5	−0.9 (−5.0, 3.1)	0.65
% Energy	13.8 (9.3, 18.2)	14.5 (10.1, 18.9)	−0.7 (−7.0, 5.6)	0.82		
Carbohydrate (g)	186.6 (125.2, 247.9)	144.3 (82.9, 205.6)	42.3 (−44.5, 129.1)	0.34		
% Energy	43.7 (29.6, 57.8)	42.3 (28.2, 56.4)	1.5 (−18.5, 21.4)	0.89	5.6 (−16.1, 27.3)	0.61
Protein (g)	70.0 (45.5, 94.6)	57.4 (32.8, 81.9)	12.7 (−22.1, 47.4)	0.47		
% Energy	16.6 (10.8, 22.5)	17.3 (11.5, 23.2)	−0.7 (−9.0, 7.6)	0.87	−2.0 (−11.1, 7.1)	0.67
Dietary fiber (g)	21.4 (14.8, 28.0)	16.4 (9.8, 23.0)	5.0 (−4.4, 14.3)	0.3	1.3 (−2.9, 5.4)	0.54
EPA ^b^ and DHA ^c^ (g)	1.6 (1.0, 2.3)	1.2 (0.6, 1.9)	0.4 (−0.5, 1.3)	0.37	0.1 (−0.4, 0.6)	0.76
Retinol (µg RE ^d^)	385.5 (261.4, 509.5)	311.0 (186.9, 435.0)	74.5 (−100.9, 249.9)	0.4	11.4 (−88.4, 111.1)	0.82
Beta carotene (µg RE)	4484.2 (2546.3, 6422.0)	3453.2 (1515.4, 5391.0)	1031.0 (−1709.5, 3771.5)	0.46	384.5 (−1936.7, 2705.8)	0.74
Vitamin C (mg)	130.7 (101.1, 160.3)	82.2 (52.6, 111.8)	48.5 (6.7, 90.4)	0.02	35.7 (5.7, 65.8)	0.02
Vitamin D (µg)	5.3 (3.3, 7.3)	4.4 (2.4, 6.4)	0.9 (−2.0, 3.7)	0.54	0.1 (−2.1, 2.3)	0.93
Niacin (mg NE ^e^)	35.8 (28.5, 43.1)	29.7 (22.4, 37.0)	6.1 (−4.3, 16.5)	0.25	1.9 (−2.2, 6.1)	0.35
Thiamin (mg)	1.3 (0.7, 1.8)	1.0 (0.4, 1.5)	0.3 (−0.5, 1.0)	0.5	−0.03 (−0.4, 0.3)	0.86
Riboflavin (mg)	1.8 (1.3, 2.4)	1.6 (1.0, 2.2)	0.2 (−0.6, 1.0)	0.55	−0.1 (−0.4, 0.2)	0.50
Vitamin B-6 (mg)	1.9 (1.3, 2.5)	1.6 (1.0, 2.2)	0.3 (−0.5, 1.1)	0.5	−0.04 (−0.4, 0.4)	0.85
Vitamin B-12 (µg)	4.2 (2.8, 5.5)	3.6 (2.3, 4.9)	0.6 (−1.3, 2.4)	0.55	−0.1 (−1.2, 1.0)	0.89
Folate (µg FE ^f^)	330.6 (215.5, 445.7)	261.0 (145.9, 376.1)	69.6 (−93.1, 232.4)	0.4	3.8 (−57.6, 65.1)	0.90
Calcium (mg)	929.7 (645.4, 1213.9)	787.9 (503.6, 1072.2)	141.8 (−260.2, 543.8)	0.49	−19.4 (−179.2, 140.4)	0.81
Iron (mg)	11.0 (6.7, 15.2)	8.7 (4.5, 13.0)	2.2 (−3.8, 8.2)	0.47	−0.2 (−2.5, 2.1)	0.85
Zinc (mg)	9.3 (6.0, 12.6)	7.6 (4.3, 10.9)	1.7 (−3, 6.3)	0.48	−0.2 (−2.0, 1.6)	0.80
Almond protein (g)	1.1 (0.5, 1.7)	1.1 (0.5, 1.7)	0.1 (−0.8, 0.9)	0.9	−0.1 (−0.9, 0.7)	0.85
Cashew protein (g)	0.6 (0.4, 0.9)	0.4 (0.2, 0.7)	0.2 (−0.1, 0.6)	0.23	0.2 (−0.2, 0.6)	0.27
Egg protein (g)	3.4 (1.4, 5.4)	3.0 (1.0, 4.9)	0.5 (−2.4, 3.3)	0.75	−0.2 (−2.6, 2.2)	0.87
Milk protein (g)	16.7 (10.8, 22.7)	15.7 (9.7, 21.6)	1.0 (−7.4, 9.4)	0.81	−1.9 (−7.1, 3.4)	0.49
Peanut protein (g)	3.9 (1.5, 6.2)	2.8 (0.4, 5.2)	1.1 (−2.3, 4.4)	0.53	0.5 (−2.6, 3.6)	0.77
Sesame protein (g)	0.2 (0.1, 0.3)	0.1 (−0.01, 0.2)	0.1 (−0.1, 0.2)	0.33	0.1 (−0.1, 0.2)	0.42
Walnut protein (g)	0.1 (−0.03, 0.1)	0.1 (−0.02, 0.14)	0 (−0.1, 0.1)	0.96	−0.01 (−0.1, 0.1)	0.88
Overall MDI ^g^ (A.U.)	73.3 (50.1, 96.5)	73.4 (50.1, 96.6)	−0.1 (−32.9, 32.8)	>0.99	−13.2 (−26.4, −0.1)	0.05

^a^ Bonferroni adjusted significance levels = 0.0063 for macronutrients, 0.0036 for micronutrients, and 0.007 for allergens. ^b^ EPA = eicosapentaenoic acid. ^c^ DHA = docosahexaenoic acid. ^d^ RE = retinol equivalents. ^e^ NE = niacin equivalents. ^f^ FE = folate equivalents. ^g^ MDI = Maternal Diet Index.

## Data Availability

Data availability can be requested from carina.venter@childrenscolorado.org. The data are not publicly available due to personalized information included in the dataset.

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
