# Peer review of "Validation of a Food Frequency Questionnaire: VioScreen-Allergy"

_nutrients, 2024, doi:10.3390/nu16213772_

Round 1
Reviewer 1 Report
Comments and Suggestions for Authors
Authors tested reliability and validity of an online pictorial food frequency questionnaire, which included intake of allergens (Vio-Screen Allergy).
The study is interesting even if it if it has important limitations that the authors themselves recognize: a small sample size, the majority of participants were highly educated and had a high household income.
Author Response
We would like to thank the reviewers for their thoughtful comments. We have responded to each question. Our response is in italic and any changes we made to the text are highlighted in red.
Reviewer 1
- Authors tested reliability and validity of an online pictorial food frequency questionnaire, which included intake of allergens (Vio-Screen Allergy). The study is interesting even if it if it has important limitations that the authors themselves recognize: a small sample size, the majority of participants were highly educated and had a high household income.
Response: Thank you for your positive comments. You are correct and we have highlighted these limitations in the text.
Reviewer 2 Report
Comments and Suggestions for Authors
The topic of the work conducted by Pickett-Nairne is quite relevant. However, I found some limitations that should be addressed before it can be considered for publication in Nutrients.
The authors should provide in the abstract some directions for further investigations and practical implications of their study.
I suggest the inclusion of a flowchart in the methodologies section with all the steps taken to conduct the present work. This will facilitate the readers’ comprehension.
Further details should be provided regarding the Veggie Meter® Biomarker (section 2.4) and the Maternal Diet Index (section 2.5).
How did you find your sample size adequate? Can you please justify this in the manuscript? It seems very small to my understanding and difficult to represent the study population.
I miss further discussions with the published literature from other regions. I encourage the author to deepen discuss their results with some other studies conducted in other regions.
As I previously mentioned regarding the abstract, the conclusions should be aligned with this. Some directions for further investigations and practical implications of the study should be provided.
Author Response
We would like to thank the reviewers for their thoughtful comments. We have responded to each question. Our response is in italic and any changes we made to the text are highlighted in red.
Reviewer 2
The topic of the work conducted by Pickett-Nairne is quite relevant. However, I found some limitations that should be addressed before it can be considered for publication in Nutrients.
Response: Thank you, we have addressed your comments below to the best of our ability.
- The authors should provide in the abstract some directions for further investigations and practical implications of their study.
Response: We have now added the following to the abstract (line 35 – 38): The VioScreen questionnaire can also be used in future studies to assess macro- and micronutrient intake. Additional validation studies assessing different portion sizes and foods eaten by infants and young children are currently undergoing.
- I suggest the inclusion of a flowchart in the methodologies section with all the steps taken to conduct the present work. This will facilitate the readers’ comprehension.
Response: We have now added the following table to the text:
Table 1. Overview of Study Procedures
|
|
Baseline Visit (Week 1) |
Week 2 |
Week 3 |
Week 4 |
Week 5 |
Week 6 (End of Study) |
|
Visit Location |
Clinic |
Phone |
Phone |
Phone |
Phone |
Clinic |
|
Informed Consent |
X |
|
|
|
|
|
|
Baseline questionnaire |
X |
|
|
|
|
|
|
VioScreen questionnaire |
X |
|
|
|
|
X |
|
Veggie Meter® |
X |
|
|
|
|
X |
|
24-hour dietary recall |
|
X* |
X^ |
X* |
X^ |
|
* Weekend dietary recall
^ Weekday dietary recall
- Further details should be provided regarding the Veggie Meter® Biomarker (section 2.4) and the Maternal Diet Index (section 2.5).
Response: We have added the following information to the text.
(line 126 - 127) The Veggie Meter® is a non-invasive portable device that measures skin carotenoid levels, a concentration biomarker of usual fruit and vegetable intake [15].
(line 134-145) The MDI [17] was computed using data regarding maternal diet during pregnancy. The index includes dietary components associated with allergy prevention (yogurt, vegetables), and dietary components associated with an increased risk of allergy (red meat, rice/grains, fried potatoes, cold cereals, 100% fruit juice). Higher MDI scores indicate a more allergy preventive diet (increased intake of yogurt and vegetables; reduced intake of red meat, rice/grains, fried potatoes, cold cereals, 100% fruit juice), while lower MDI scores represent a less allergy preventive diet (reduced intake of yogurt and vegetables; increased intake of red meat, rice/grains, fried potatoes, cold cereals, 100% fruit juice). The theoretical range of MDI scores is 0 to 100 [17]. Coding for the MDI is freely available for download from https://github.com/CarinaVenter/MaternalDietIndex. The code allows a researcher to compute the maternal diet index using either SAS (SAS Institute,
Cary NC) or R28 statistical software.
- How did you find your sample size adequate? Can you please justify this in the manuscript? It seems very small to my understanding and difficult to represent the study population.
Response: We agree with these comments from the reviewer. The following text appears in the discussion.
(line 299 - 304) Limitations of our study include a small sample size, which may have attenuated power. However, careful examination of the mean differences in intake observed between and within the methods suggests that the differences are both clinically and statistically insignificant. Generalizability may be limited as the majority of participants were highly educated and had a high household income.
- I miss further discussions with the published literature from other regions. I encourage the author to deepen discuss their results with some other studies conducted in other regions.
Response: To our knowledge, the only other paper published looking the validation of the Vioscreen questionnaire was published by Kristal et al. [12] and we have referenced this paper in the text. We have also referenced the previous studies comparing dietary intake with the Veggie Meter outcomes.[22][23]
We have also added the following text to the paper: [Suggested a revision of this paragraph in the manuscript.]
(line 248 - 260) The validated VioScreen questionnaire has been used in a wide range of studies focusing on dietary intake in individuals suffering from e.g. cancer,[19, 20] pancreatitis,[21, 22] Parkinson’s disease,[13], dietary intake during pregnancy,[23] and studying healthy eating patterns in individuals without disease.[24, 25] A systematic review indicated that skin carotene measures are valid measures of assessing fruit and vegetable intake.[26] The Veggie Meter® and assessment of skin carotene levels have been used in studies assessing dietary intake, [27-29] and studying associations between dietary intake in health outcomes.[30-32] The Vioscreen questionnaire has not previously been used in studies focusing on allergy outcomes, we plan to do in future. The Veggie Meter® has previously been used to assess carotene intake in children with cow’s milk allergy. The authors reported significantly higher levels of skin carotenoids in children with cow’s milk allergy compared to controls.[33]
- As I previously mentioned regarding the abstract, the conclusions should be aligned with this. Some directions for further investigations and practical implications of the study should be provided.
Response: We have now added the following to the conclusion (line 311-314): The VioScreen questionnaire can also be used in future studies to assess macro- and micronutrient intake. Additional validation studies assessing different portion sizes and foods eaten by infants and young children are currently undergoing.
Round 2
Reviewer 2 Report
Comments and Suggestions for Authors
I'm satisfied with the answers provided and the improvements made.